# An Interval AHP Technique for Classroom Teaching Quality Evaluation

Ya Qin [1,2], Siti Rahayu Mohd. Hashim [1,*] and Jumat Sulaiman [1]

1 Faculty of Science and Natural Resources, Universiti Malaysia Sabah, Kota Kinabalu 88400, Sabah, Malaysia
2 School of Mathematics and Information Sciences, Neijiang Normal University, Neijiang 641000, China
* Correspondence: rahayu@ums.edu.my

**Abstract:** Classroom teaching evaluation is one of the most important ways to improve the teaching quality of mathematics education in higher education, and it is also a group decision making problems. Meanwhile, there is some uncertain information in the process of evaluation. In order to deal with this uncertainty in classroom teaching quality evaluation and obtain a reliable and accurate evaluation result, an interval analytic hierarchy process (I-AHP) is employed. To begin with, the modern evaluation tool named RTOP is adapted to make it more consistent with the characteristics of the discipline. In addition, the evaluation approach is built by using the I-AHP method, and some details of weights of the criteria and assessors are developed, respectively. Thirdly, a case study has been made to verify the feasibility of the assessment approach for classroom teaching quality evaluation on mathematics. Finally, a comprehensive evaluation of classroom quality under an interval number environment is conducted, and some results analyses and comparisons are also discussed to show that the proposed approach is sound and has a stronger ability to deal with uncertainty.

**Keywords:** RTOP; classroom teaching quality evaluation; interval analytic hierarchy process; comprehensive evaluation

## 1. Introduction

Classroom teaching quality evaluation is an important link and content of teaching quality management in colleges and universities. However, how to make a quantitative and comprehensive evaluation of teachers' classroom teaching quality is indeed difficult and worthy of study. There are many indexes that should be considered in the evaluation design, and it is very difficult to completely eliminate the deviation of the evaluation indexes due to the knowledge level, cognitive ability and personal preference of the evaluators, that is, there are some uncertainties in evaluation although the indexes have qualitative descriptions. Therefore, classroom teaching quality evaluation is a multi-objective decision making problem.

The analytic hierarchy process (AHP) [1] is a practical multi-objective decision making method, which was put forward by Saaty in the 1980s. AHP's main characteristic is that it reasonably combines the qualitative and quantitative decision making. AHP decomposes the decision making problem into different hierarchical structures according to the order of the general objective, sub-objectives of each level, evaluation criteria and specific alternative investment scheme, and then use the method of solving the eigenvector of the judgment matrix to obtain the priority weight of each element to a certain element of the upper level, and finally use the method of weighted sum to merge the final weight of each alternative scheme to the general objective in a hierarchical manner. The best scheme is the one with the largest final weight. AHP is more suitable for the decision making problem of the target system with layered and staggered evaluation indexes, and the target value is difficult to describe quantitatively. AHP has the advantages of systematization, conciseness and practicality, and less quantitative data information, so it is widely used in various

comprehensive evaluation problems. In general, the studies of AHP are mainly focused on the following three aspects: the consistency of the judgment matrix, new methods that are developed to address the consistency of the judgment matrix and studies on the scaling of AHP.

The consistency of the judgment matrix. Liang [2] and Ma [3] improved the traditional AHP by combining the optimal transfer matrix. The advantage of this method is that it does not need to carry out the consistency test. However, they only made local corrections to the judgment matrix. From the global perspective, Jin [4] put forward an accelerated genetic algorithm to modify the consistency of the judgment matrix and calculate the ranking weight. Wang [5] analyzed the causes of an inconsistent judgment matrix, and Li [6] proposed a multi-attribute variable weight decision-making method based on the BG-AHP. This method can reflect the real preference of decision makers, weaken the error caused by subjective judgment of decision makers, and finally obtain an accepted conclusion. In the above study, the researchers adjusted some elements to meet the consistency of the judgment matrix, but after the adjustment, the original judgment information was tampered with, so the reliability of the conclusion was reduced. Under this background, Wang [7] proposed a ranking method of a non-uniformity judgment matrix based on manifold learning.

A series of new methods are developed to address the consistency of the judgment matrix and group decision making. Wei et al. [8,9] improved the consistency of the judgment matrix by adjusting the elements in the judgment matrix. Wei et al. [8] modified a pair of elements of the judgment matrix based on the existing consistency test standard to improve the consistency of the judgment matrix. Meanwhile, Zhu et al. [9] adjusted the elements of the matrix by measuring the distance between each element of the judgment matrix and its value when it reaches the best consistency. Tian [10] combined the possible satisfaction index and consistency ratio standard to control the improvement direction and adjustment strength of the judgment matrix. The consistency test and modification of the judgment matrix are the key steps of group decision making. Sun [11] introduced possible satisfaction into a new algorithm for improving the compatibility of incomplete matrix and ranking. A novel aggregation approach for AHP judgment matrices was introduced to solve the group decision problem [12].

Studies on the scaling of AHP. Liu [13,14] elaborated on the basic principle, basic steps and calculation methods of AHP. He [15] compared the ranking results under different scales and emphasized that the group judgment scale system has an important impact on the reliability of the results of AHP. Based on Xu [16], Wang [17], Hou [18] and Luo [19], various scale methods and existing scale comparison studies were proposed, various performance evaluation standards were established, several common scale methods were compared and analyzed, and reference scales for different ranking problems were proposed [20].

Traditional AHP replaces the absolute scale with the relative scale, and makes full use of people's experience and judgment ability. The scale is an integer between 1 and 9 and is reciprocal, which is in line with people's psychological habits when making judgment. However, there are many uncertainties in decision making problems, such as the preferences of experts, etc. So, the integer scale is no longer suitable for describing this kind of uncertainty; on the contrary, the interval scale is more suitable for the judgment of uncertainty than the integer scale. Therefore, Wu [21] proposed an extension of AHP named interval AHP (I-AHP), in which the judgment matrix is given by the interval judgment matrix. I-AHP is an improvement on traditional AHP. In the process of establishing a pair of judgment matrices, interval numbers are used to replace single point values. This can reduce the influence of human subjective will and better reflect the uncertainty of judgment. Since the appearance of I-AHP, it has attracted more attention and has been applied to the social–economic system successfully. Deng [22] used the I-AHP method to establish the structural hierarchy and applied it to China railway track system (CRTS) III's prefabricated slab track cracking condition. Milosevic [23] studied the sustainable management for the architectural heritage in smart cities by using fuzzy and I-AHP methods. Wang [24,25]

proposed an improved interval AHP method and hybrid interval AHP-entropy method for assessment of a cloud platform-based electrical safety monitoring system. Ghorban-zadeh [26] built an I-AHP group decision support model for sustainable urban transport planning considering different stake holders. Moslem [27] analyzed stakeholder consensus for a sustainable transport development decision by the fuzzy AHP and I-AHP.

The determination of the evaluation index weight has always been a core issue of evaluation, and teaching quality evaluation is no exception. However, traditional AHP has some shortcomings in determining the weight of evaluation indicators, mainly in the following two aspects: (1) the comprehensive evaluation of classroom teaching quality is a complex evaluation process, which requires the experience and professional knowledge of assessors to compare and judge the weight of influencing factors. However, due to the difference of assessors' experience and expertise, the credibility of the judgment matrix given by each assessor is often different. (2) Traditional AHP is used to construct the judgment matrix. The assessors compare the influencing factors in pairs, and the result is a definite integer solution. However, referring to the uncertainties and complexities of the influencing factors, this kind of accurate numerical judgment is not so "accurate". Instead, it needs to use "fuzzy" interval judgment to reflect the judgment conclusion. Therefore, this work is based on the theory of fuzzy mathematics. Considering the difference of assessor evaluation and the uncertainty of weight determination. The interval AHPs are developed further to determine the weight of the indexes to obtain a more reasonable and accurate comprehensive evaluation method of classroom teaching quality.

Based on the above goals, this paper is arranged as follows: Some basic concepts on interval numbers and the judgment matrix are reviewed in Section 2; meanwhile, the AHP method and the evaluation tool are introduced in this section. In Section 3, the classroom teaching quality evaluation approach is built by applying the I-AHP method. A case study is carried out in Section 4 based on the proposed approach in Section 3. A comprehensive evaluation of classroom quality under an interval number environment is conducted in Section 5. Some results analyses and comparisons are discussed in Section 6, and conclusions are made in Section 7. The structure of this work can be described in Figure 1.

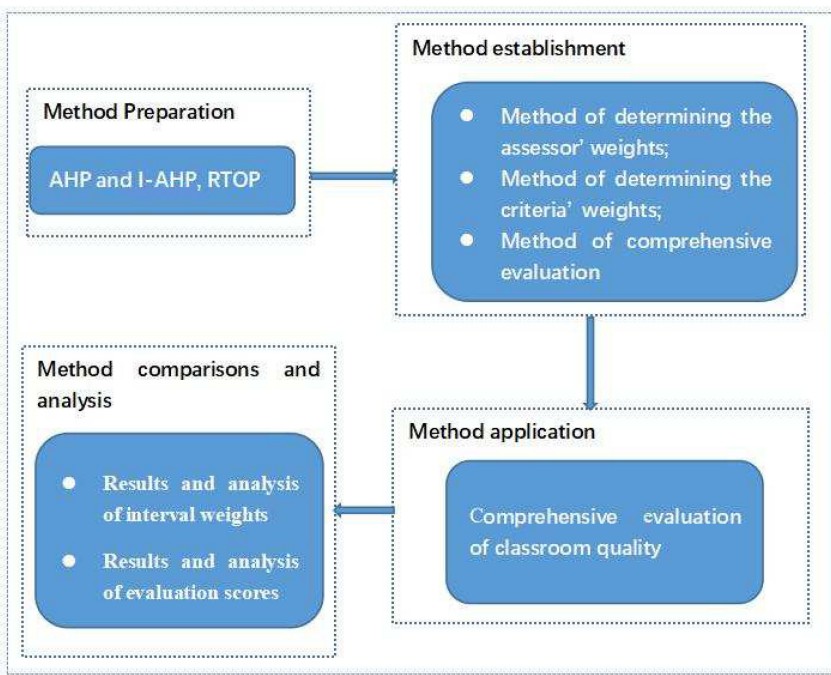

**Figure 1.** The structure of this paper.

## 2. Preliminaries

In this section, the basic concepts of the interval number and judgment matrix will be introduced. Then, a brief explanation on the implementation process of AHP will be given. Finally, the hierarchical evaluation tool, Reformed Teaching Observation Protocol (RTOP), will be introduced and discussed briefly.

### 2.1. Interval Number and Interval Judgment Matrix

An interval number can be expressed as a real interval $v = [v^-, v^+]$, where $v^- \leq v^+$ and $v^-, v^+ \in R$. For any two interval numbers $v_1 = [v_1^-, v_1^+]$ and $v_2 = [v_2^-, v_2^+]$, $v_1 = v_2$ if and only if $v_1^- = v_2^-, v_1^+ = v_2^+$.

**Definition 1.** *[28] For any two interval numbers $v_1 = [v_1^-, v_1^+]$ and $v_2 = [v_2^-, v_2^+]$, then*
*(1) $v_1 + v_2 = [v_1^- + v_2^-, v_1^+ + v_2^+]$;*
*(2) $v_1 v_2 = [v_1^- v_2^-, v_1^+ v_2^+]$;*
*(3) $kv_1 = [kv_1^-, kv_1^+]$;*
*(4) $\frac{v_1}{v_2} = \left[\frac{v_1^-}{v_2^+}, \frac{v_1^+}{v_2^-}\right]$, especially, $\frac{1}{v_1} = \left[\frac{1}{v_1^+}, \frac{1}{v_1^-}\right]$.*

**Definition 2.** *[28] $\Lambda = (c_{ij})_{n \times n}$ is called an interval judgment matrix, if for all $i, j \in \{1, 2, \cdots, n\}$, we have*

$$c_{ij} = \left[c_{ij}^-, c_{ij}^+\right],$$

*where $1/9 \leqslant c_{ij}^- \leqslant c_{ij}^+ \leqslant 9$ and $c_{ji} = 1/c_{ij}$.*

### 2.2. Analytic Hierarchy Process

In dealing with multi-criteria decision making (MCDM) problems, assessors can assign the weight of each evaluation criteria. However, this is a difficult issue because each assessor has different opinions or preferences about the importance of the evaluation criteria, which could create conflict with the evaluation objective. As one of the most popular MCDM methods, the AHP method could be used to overcome this obstacle [29]. The AHP method is a technique for deriving ratio scales from paired comparisons, which decomposes the elements that are always related to decision-making into objectives, factors, criteria and other levels and then conducts qualitative and quantitative analysis on this basis [1]. Its solution steps are mainly as follows: (1) Establishing a hierarchical structure model. In this model, the decision-making objects are divided into objective level, criteria level, factor level and indicator level according to their mutual relations. (2) Constructing a pairwise comparison judgment at each level. Based on the AHP model, the assessors are required to compare the importance of a pair of factors. This research adopts Saaty's 1–9 scale (as shown in Table 1) of importance to rate the scale of importance of the given factor. (3) Calculating the matrix weight (eigenvector). When the assessors make comparison judgments, they need to consider the consistency between different evaluation criteria (such as $F_i$, $F_j$ and $F_k$). For example, if the ratio of the importance of index $F_i$ and $F_j$ is 3 ($F_i = 3F_j$), and the ratio of the importance of index $F_j$ and $F_k$ is 4 ($F_j = 4F_k$), then the ratio of the importance of index $F_i$ and $F_k$ is 12 ($F_i = 12F_k$) under a completely consistent judgment. However, people's judgments cannot be completely consistent. In an actual MCDM process, the pairwise comparisons may be judged as $F_i = 3F_j$, $F_j = 4F_k$ and then $F_i = 11F_k$. Obviously, this result is still reasonable and logical. Considering some inconsistency is inevitable in human judgment, the AHP method allows a small degree of internal inconsistencies. However, the inconsistencies need to be controlled within an allowable range, and the common threshold is 0.1 (that is, not more than 0.1) [30]. The following steps further explain the issues related to the consistency test. (4) Calculating the maximum eigenvalue, (5) computing the consistency index with an eigenvalue. (6) Measuring the consistency ratio with CI and random index. If the value of the CR is less than 0.1, the data of the judgment matrix is reliable, and the judgment matrix is considered to have a reliable consistency [31].

Otherwise, the judgment matrix needs to be adjusted until it passes the consistency test. The last step is (7) making decisions with the obtained results.

**Table 1.** AHP scale of importance (Saaty,2008,1990).

| Value of Importance | Comparative Judgment |
|---|---|
| 1 | $F_i$ is as important as $F_j$ |
| 3 | $F_i$ is slightly more important than $F_j$ |
| 5 | $F_i$ is strongly more important than $F_j$ |
| 7 | $F_i$ is very strongly more important than $F_j$ |
| 9 | $F_i$ is extremely more important than $F_j$ |
| 2,4,6,8 | Represents the median value of the above adjacent judgment |
| Reciprocal | If the ratio of the importance of $F_i$ and $F_j$ is $f_{ij}$, then ratio of $F_j$ and $F_i$ is $f_{ji} = 1 \big/ f_{ij}$ |

*2.3. Hierarchical Evaluation Structure: Reformed Teaching Observation Protocol*

In order to evaluate teachers' teaching performance comprehensively and accurately, relevant theoretical and empirical studies have shown that the classroom teaching quality evaluation system should be designed from diverse angles and aspects: teaching attitude, teaching preparation, teaching process, teaching content, etc. [32–35]. However, it is quite difficult and complex to design a reasonable and scientific teaching quality evaluation system [36–38], because a series of quantitative analyses and modern tests of reliability and validity are required.

This study chose a classroom observation instrument called "Reformed Teaching Observation Protocol (RTOP)" as the evaluation tool. This instrument was proposed to constructively critique details of classroom practices (cooperative learning, interactive engagement, etc.), capture the current reform movement and improve the preparation of science and mathematics teachers by the ACEPT (Arizona Collaborative for Excellence in the Preparation of Teachers) evaluation team at Arizona State University in 1995 [39–41]. The RTOP consists of five factors: lesson design and implementation (short for $F_1$), content—propositional knowledge ($F_2$), content—procedural knowledge ($F_3$), classroom culture—communicative interactions ($F_4$), and classroom culture—student/teacher relationship ($F_5$). Each factor contains five observable items, and the items contribute to their corresponding factors.

The RTOP has been proven to have high reliability [42] and prediction validity [43] after a long-term strict development process and experimental data analysis. Furthermore, the RTOP was mentioned as having multiple positive effects. For students, the RTOP was found to have an association with prominent student-centered active learning increases in science and mathematics courses [44–46]. For both new and veteran teachers working with RTOP, it was found that the RTOP is useful not only for achieving teaching purposes, scoring their own teaching, but more importantly for acquiring insight into their own teaching practices that guides their instructional improvement and professional teaching growth. The RTOP has also been used to evaluate the effectiveness of professional development programs [44–46], for course design [47], as a peer evaluation tool [46], and as a standard to construct the concurrent validity of newer observation tools [48,49]. Hence, RTOP is regarded and accepted as a mature and professional classroom teaching evaluation instrument that conforms to modern educational ideas, and correspondingly would contribute to evaluating and improving teaching effectiveness in higher education. Therefore, the RTOP model is suitable to evaluate mathematics courses in higher education. RTOP's items could be further refined, and the scoring rules can also be modified through discussion and consultation [50,51] under an actual situation. In this paper, some items in RTOP were revised for mathematics courses' teaching evaluation. Consequently, a three levels hierarchical structure of RTOP is shown in Table 2.

**Table 2.** A hierarchical evaluation structure of RTOP.

| Target Level | Factor Level ($F$) | Item Level ($I$) |
|---|---|---|
| Teaching quality evaluation | $F_1$. Lesson Design and Implementation | $I_1$. Respect student preconceptions and knowledge of mathematics<br>$I_2$. Form a math learning group<br>$I_3$. Explore before formal presentation<br>$I_4$. Seek alternative approaches different in textbooks<br>$I_5$. Adopt student ideas in teaching |
| | $F_2$. Content: Propositional Knowledge | $I_6$. Involve fundamental concepts of mathematics<br>$I_7$. Promote coherent understanding of mathematical concepts<br>$I_8$. Teacher have a solid grasp of the contents (especially for unrelated questions)<br>$I_9$. Encourage abstraction (mathematics models or formulas)<br>$I_{10}$. Emphasize the connection between mathematics and other disciplines or social life |
| | $F_3$. Content: Procedural Knowledge | $I_{11}$. Students use models, formulas, graphics to express their understanding<br>$I_{12}$. Students make predictions, assumptions or estimates<br>$I_{13}$. Make critical inferences or estimates of results<br>$I_{14}$. Students reflect on their learning in mathematics class<br>$I_{15}$. Students infer or question corresponding conclusions, concepts and formulas |
| | $F_4$. Classroom culture: Communicative Interactions | $I_{16}$. Students communicate their understanding and ideas with various ways<br>$I_{17}$. Teachers' questions lead to students' thinking differently about mathematics<br>$I_{18}$. Students actively discuss mathematics problems<br>$I_{19}$. The direction of the class is determined by the discussion of students<br>$I_{20}$. Students actively express their views without being ridiculed |
| | $F_5$. Classroom culture: Student/ Teacher Relationships | $I_{21}$. Encourage students to actively participate in discussion<br>$I_{22}$. Encourage students to solve mathematical problems in many ways<br>$I_{23}$. Teacher is patient when students think about problems or complete assignments<br>$I_{24}$. When students investigate or study, the teacher acts as a resource<br>$I_{25}$. Teacher listens carefully when students discuss and express their views |

## 3. Methodology

This study adopts the quantitative approach using questionnaires that consist of interval numbers, interval matrices as the main instrument, and interval AHP as the main method. Before conducting comprehensive evaluation for a course, the evaluation criteria weights and assessors' weights need to be determined.

### 3.1. Determining the Evaluation Criteria' Weights with I-AHP

3.1.1. Constructing Interval Judgment Matrices

The solution steps and methods of I-AHP are roughly the same as those of traditional AHP method. After a three-level hierarchical structure model was established with RTOP, the assessors need to compare two pairwise factors'/items' importance or preference and rate the scale of importance of the chosen factor/item with interval numbers rather than crisp numbers in traditional judgment matrices. Additionally, the interval judgment matrix is also constructed with Saaty's 1–9 comparison scale.

An interval pairwise comparison matrix consisting of interval numbers will be obtained as

$$
\Lambda = \begin{pmatrix}
[1,1] & \left[c_{12}^-, c_{12}^+\right] & \cdots & \left[c_{1n}^-, c_{1n}^+\right] \\
\left[\frac{1}{c_{12}^+}, \frac{1}{c_{12}^-}\right] & [1,1] & \cdots & \left[c_{2n}^-, c_{2n}^+\right] \\
\cdots & \cdots & \cdots & \cdots \\
\left[\frac{1}{c_{1n}^+}, \frac{1}{c_{1n}^-}\right] & \left[\frac{1}{c_{2n}^+}, \frac{1}{c_{2n}^-}\right] & \cdots & [1,1]
\end{pmatrix}
\tag{1}
$$

For further calculations of the I-AHP, matrix $\Lambda$ should be a reciprocal one, and the judgment matrix of I-AHP could be separated into two matrices: the lower bound matrix $\Lambda^-$ and the upper bound matrix $\Lambda^+$ [52] as follows:

$$
\Lambda^- = \left(c_{ij}^-\right)_{nn} = \begin{pmatrix}
1 & c_{12}^- & \cdots & c_{1n}^- \\
\frac{1}{c_{12}^-} & 1 & \cdots & c_{2n}^- \\
\cdots & \cdots & \cdots & \cdots \\
\frac{1}{c_{1n}^-} & \frac{1}{c_{2n}^-} & \cdots & 1
\end{pmatrix}
\tag{2}
$$

$$
\Lambda^+ = \left(c_{ij}^+\right)_{nn} = \begin{pmatrix}
1 & c_{12}^+ & \cdots & c_{1n}^+ \\
\frac{1}{c_{12}^+} & 1 & \cdots & c_{2n}^+ \\
\cdots & \cdots & \cdots & \cdots \\
\frac{1}{c_{1n}^+} & \frac{1}{c_{2n}^+} & \cdots & 1
\end{pmatrix}
\tag{3}
$$

Obviously, the matrices $\Lambda^-$ and $\Lambda^+$ are reciprocal. Now the matrix $\Lambda$ is also called an interval reciprocal matrix.

3.1.2. Calculating Criteria' Basic Weights

Step 1. Computing eigenvectors

According to the matrix $\Lambda^-$ ($\Lambda^+$), the feature vector of matrix $\Lambda^-$ ($\Lambda^+$) could be computed with a geometric mean (GM) method as shown below:

$$
\tilde{w}_{b_i}^- = \sqrt[n]{\Pi_{j=1}^n c_{ij}^-}
\tag{4}
$$

$$
\tilde{w}_{b_i}^+ = \sqrt[n]{\Pi_{j=1}^n c_{ij}^+}.
\tag{5}
$$

Since different evaluation criteria often have different dimensions, such a situation will affect the comparison results of the data analysis. To eliminate the dimensional influence between criteria, weights normalization is required, and the specific formula is given in Equations (6) and (7).

$$w_{b_i}^- = \tilde{w}_{b_i}^- \bigg/ \sum_{i=1}^{n} \tilde{w}_{b_i}^- \tag{6}$$

$$w_{b_i}^+ = \tilde{w}_{b_i}^+ \bigg/ \sum_{i=1}^{n} \tilde{w}_{b_i}^+ \tag{7}$$

So the eigenvectors $w_b^- = \left(w_{b_1}^-, w_{b_2}^-, \cdots, w_{b_n}^-\right)^T, w_b^+ = \left(w_{b_1}^+, w_{b_2}^+, \cdots, w_{b_n}^+\right)^T$ could be obtained.

Step 2. Calculating the maximum feature root of the judgment matrix

Taking the lower bound matrix $\Lambda^-$ as an example, the judgment matrix $\Lambda^-$ could be calculated with Equation (8).

$$\lambda_{\max} = \frac{1}{n} \sum_{i=1}^{n} \frac{\left(\Lambda^- w_b^-\right)_i}{w_{b_i}^-} \tag{8}$$

Step 3. Calculating the judgment matrix's consistency index

The consistency index is used to examine the consistency of assessors' judgment thinking of setting weights with Equation (9).

$$CI = \frac{\lambda_{\max} - n}{n - 1} \tag{9}$$

in which, $n$ is the number of rows in corresponding matrix.

Step 4. Determining the consistency ration of the judgment matrix $\Lambda^-$ with Equation (10).

$$CR = CI/RI \tag{10}$$

where random consistency index presents the average ratio index of the judgment matrix. The RI values is computed under different sizes of matrices by Saaty, which is shown in Table 3.

**Table 3.** Satty's derivation of consistency index for a randomly generated matrix.

| n | 1 | 2 | 3 | 4 | 5 | 6 | 7 | 8 | 9 | 10 | 11 | 12 |
|---|---|---|---|---|---|---|---|---|---|----|----|----|
| RI | 0.00 | 0.00 | 0.58 | 0.90 | 1.12 | 1.24 | 1.32 | 1.41 | 1.45 | 1.49 | 1.51 | 1.48 |

When the value of $CR < 0.1$, it is considered that the matrix $\Lambda^-$ is consistent; that is, the index's weight gained by the assessors' comparison is reasonable, and the results obtained by I-AHP are scientific and effective. Otherwise, the judgment matrix needs to be modified or reconstructed until the qualified consistency requirement is gained to ensure the rationality of decisions.

Similarly, the upper bound judgment matrix $\Lambda^+$ should also be measured via a consistency test. If both the lower bound judgment matrix $\Lambda^-$ and the upper bound judgment matrix $\Lambda^+$ have qualified consistency ($CRs < 0.1$), then the interval matrix $\Lambda$ also has qualified consistency, namely the $CR$ of matrix $\Lambda$ will be less than 0.1. Otherwise, the matrix $\Lambda$ does not passes the consistency test [52,53].

In this case, the weight vector of the interval pairwise comparison matrix $\Lambda$ could be calculated from the following Equation (11) [54].

$$w_b = [w_{bL}, w_{bR}] = \left[\alpha w_b^-, \beta w_b^+\right], \tag{11}$$

In which,

$$\alpha = \left[ \sum_{j=1}^{n} \left( 1 \bigg/ \sum_{i=1}^{n} c_{ij}^{+} \right) \right]^{1/2}, \beta = \left[ \sum_{j=1}^{n} \left( 1 \bigg/ \sum_{i=1}^{n} c_{ij}^{-} \right) \right]^{1/2} \tag{12}$$

Therefore, criteria' basic weights have been obtained as Equation (11).

### 3.2. Determining the Assessor' Weight

Obviously, in decision-making, it is not very scientific to give the same weight to different assessors due to their different research experience, preferences and profession knowledge [55]. Therefore, it is necessary to gain an objective weight of assessors. This study combines similarity and difference principles to determine the weights of the assessors' judgments. To avoid repetitive and complicated computation, the assessors' weights will be obtained from the interval judgment matrices in the target level; that is, factors $F_1$–$F_5$.

### 3.2.1. Calculating the Similarity Coefficient of Assessors' Evaluation

In this section, all vectors and matrices are discussed under real numbers. For any two vectors $a^k = \left( a_1{}^k, a_2{}^k, \cdots, a_n{}^k \right)$ given by assessor-$k$, $b^l = \left( b_1{}^l, b_2{}^l, \cdots, b_n{}^l \right)$ given by assessor-$l$ ($k, l = 1, 2, \cdots, m$), a cosine similarity measure between $a^k$ and $b^l$ is shown as Equation (13) [56]:

$$s_{kl} \left( a^k, b^l \right) = \frac{\sum\limits_{i=1}^{n} a_i^k b_i^l}{\sqrt{\sum\limits_{i=1}^{n} \left( a_i^k \right)^2} \sqrt{\sum\limits_{i=1}^{n} \left( b_i^l \right)^2}} \tag{13}$$

The sum of the similarity evaluated by assessor-$k$ is

$$s_k^{'} = \sum_{l=1}^{m} s_{kl} - 1 \tag{14}$$

After normalization, the similarity of assessor-$k$ will be

$$S^k = s_k^{'} \bigg/ \sum_{l=1}^{m} s_k^{'}. \tag{15}$$

Obviously, with a larger $S^k$, the evaluation of assessor-$k$ will be closer to others' evaluations, so the weight of assessor-$k$ will be greater.

### 3.2.2. Calculating the Difference of Assessors' Evaluation

Similarly, for any judgment matrix that can be expressed as the form of block matrix $\Psi_j = \left( \psi_{1j}, \psi_{2j}, \cdots, \psi_{nj} \right)^T$, then the average evaluation values for item-$i$ from $m$ assessors can be calculated as Equation (16) [57]:

$$\delta_i = \sum_{j=1}^{m} \psi_{ij} \bigg/ m \tag{16}$$

Again, for item-$i$, the difference value between the values of assessor-$k$ and the average values $\delta_i$ will be

$$\sigma_{ik} = |\psi_{ik} - \delta_i| \tag{17}$$

Set $\sigma_k = \sum\limits_{i=1}^{n} \sigma_{ik}$, the degree of difference of assessor-$k$ will be gained as

$$D^k = \sigma_k \Big/ \sum_{k=1}^{m} \sigma_k \tag{18}$$

### 3.2.3. Calculating the Weight of Evaluation Assessors

Let the weight of assessor-$k$ be $\omega_{a_k}$, then

$$\omega_{a_k} = \begin{cases} \dfrac{S^k(1-D^k)}{1-\sum\limits_{k=1}^{n} S^k D^k} , & \text{if } \sum\limits_{k=1}^{n} S^k D^k \neq 1; \\[4mm] S^k , & \text{if } \sum\limits_{k=1}^{n} S^k D^k = 1. \end{cases} \tag{19}$$

### 3.2.4. Calculating the Criteria' Final Weights

According to the operation rules of interval numbers, the final criteria weights could be aggregated with the basic criteria weights and assessors' weights as Equation (20):

$$w = [w^-, w^+] = \left[ \sum_{i=1}^{n} \omega_{a_i}^- w_{b_i}^-, \sum_{i=1}^{n} \omega_{a_i}^+ w_{b_i}^+ \right]. \tag{20}$$

### 3.3. Comprehensive Evaluation Model

According to the above analysis, the evaluation procedure can be summarized as following Figure 2.

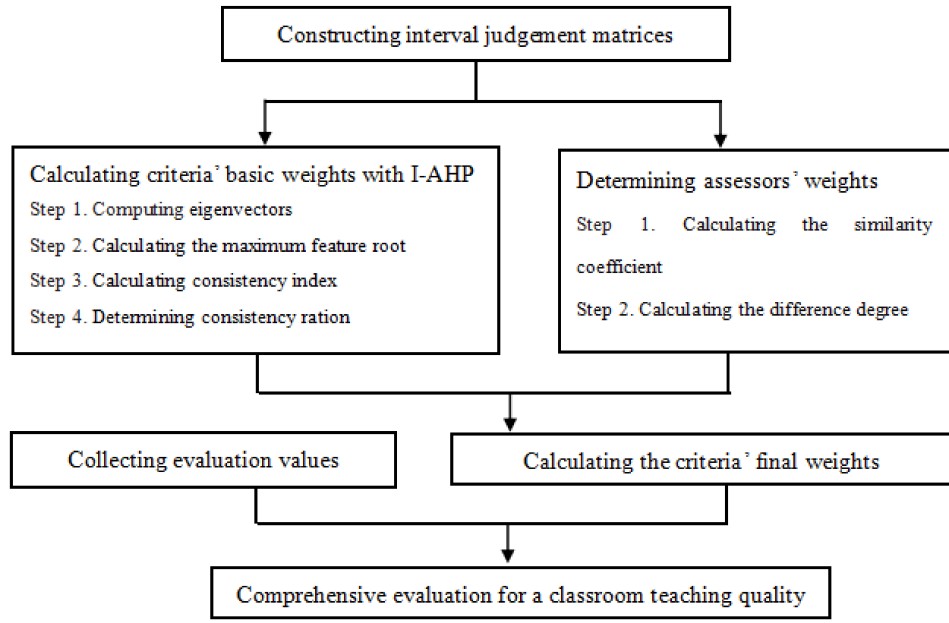

**Figure 2.** Evaluation framework on classroom teaching quality evaluation.

## 4. Case Study

Neijiang Normal University is a local public university. It has always attached importance to classroom teaching quality and cultivation of high-quality students. In mathematics teacher education, the statistics course is a mathematics subject that studies and reveals the statistical regularity of random phenomena, and which is a professional basic course for mathematics and applied mathematics majors. From this course, learners could obtain

competencies including systems thinking, anticipatory and critical thinking, and cooperation ability [58]. Such competencies are key factors for achieving sustainable development for students and the necessity of teaching for teachers.

### 4.1. Evaluation Object and Subject

This study chooses a classroom teaching video as the evaluation object. The video course is named "Bayesian formula and its application" in ≪ Probability Theory and Mathematical Statistics≫ from teacher L in Neijiang Normal University. Teacher L participated in the "First young teachers' lecture competition" and has won the first prize.

As to the evaluation subjects, this study invites five assessors with mathematics backgrounds: an expert, a colleague, an administrator, a student and teacher L himself.

The purpose of the following section is to obtain the basic criteria weights and assessors' weights with the methods in Section 3.

### 4.2. Constructing Interval Judgment Matrices

For the factor levels $F_1$ to $F_5$, interval reciprocal judgment matrices $\Lambda_i (i = 1, 2, \cdots, 5)$ from five different assessors are listed as follows:

$$\Lambda_1 = \begin{pmatrix} [1,1] & [3,4] & [4,5.5] & [0.5,2] & [4,5] \\ & [1,1] & [1.5,2] & [0.2,0.5] & [2,3.5] \\ & & [1,1] & [0.2,1/3] & [0.5,1] \\ & & & [1,1] & [2,3] \\ & & & & [1,1] \end{pmatrix},$$

$$\Lambda_2 = \begin{pmatrix} [1,1] & [1/3,1] & [0.5,0.8] & [2,3] & [1,4] \\ & [1,1] & [1,2] & [3,5] & [2,3] \\ & & [1,1] & [3,4.5] & [2,2.5] \\ & & & [1,1] & [2,3] \\ & & & & [1,1] \end{pmatrix},$$

$$\Lambda_3 = \begin{pmatrix} [1,1] & [3,4.5] & [2,3] & [0.5,0.8] & [1/3,1] \\ & [1,1] & [0.5,0.8] & [1/3,0.5] & [0.5,1] \\ & & [1,1] & [0.25,0.5] & [2/7,0.75] \\ & & & [1,1] & [2.5,3] \\ & & & & [1,1] \end{pmatrix},$$

$$\Lambda_4 = \begin{pmatrix} [1,1] & [1/3,0.75] & [0.2,0.4] & [1/3,0.5] & [0.5,0.8] \\ & [1,1] & [0.5,0.75] & [1/3,2/3,] & [0.5,0.6] \\ & & [1,1] & [2,3.5] & [3,4] \\ & & & [1,1] & [3,3.5] \\ & & & & [1,1] \end{pmatrix},$$

$$\Lambda_5 = \begin{pmatrix} [1,1] & [3,4] & [2,2.5] & [1,2] & [1,1.8] \\ & [1,1] & [3,4] & [2,2.5] & [1.5,2.2] \\ & & [1,1] & [0.5,0.5] & [0.5,0.8] \\ & & & [1,1] & [1.5,2] \\ & & & & [1,1] \end{pmatrix}.$$

Taking the interval judgment matrix $\Lambda_1$ as an example for calculation, $\Lambda_1$ can be divided into the lower bound matrix $\Lambda_1^-$ and the upper bound matrix $\Lambda_1^+$:

$$\Lambda_1^- = \begin{pmatrix} 1 & 3 & 4 & 1/2 & 4 \\ 1/3 & 1 & 3/2 & 1/5 & 2 \\ 1/4 & 2/3 & 1 & 1/5 & 1/2 \\ 2 & 5 & 2 & 1 & 2 \\ 1/4 & 1/2 & 2 & 1/2 & 1 \end{pmatrix}$$

and

$$\Lambda_1^+ = \begin{pmatrix} 1 & 4 & 11/2 & 2 & 5 \\ 1/4 & 1 & 2 & 1/2 & 7/2 \\ 2/11 & 1/2 & 1 & 1/3 & 1 \\ 1/2 & 2 & 3 & 1 & 3 \\ 1/5 & 2/7 & 1 & 1/3 & 1 \end{pmatrix},$$

respectively.

*4.3. Calculation the Weights of Factors $F_1$–$F_5$*

In this subsection, we take the factor $F_1$ as an example to show the determined procedure of weight, and the weights of $F_2$–$F_5$ can be obtained similarly.

4.3.1. Calculation the Basic Weights of Factors $F_1$–$F_5$

Step 1. Calculating the feature vector with Equations (4)–(7). The feature vector $\tilde{w}_{b_1}^-, \tilde{w}_{b_1}^+$ of matrix $\Lambda_i^-, \Lambda_i^+$ are listed as follows:

$$\tilde{w}_{b_1}^- = (1.8957, 0.7248, 0.4409, 2.5020, 0.6598)^T \tag{21}$$

$$\tilde{w}_{b_1}^+ = (2.9409, 0.9736, 0.4969, 1.5518, 0.4529)^T \tag{22}$$

After normalization, the feature vector will be

$$w_{b_1}^- = (0.3046, 0.1165, 0.0709, 0.4020, 0.1060)^T \tag{23}$$

$$w_{b_1}^+ = (0.4584, 0.1517, 0.0774, 0.2419, 0.0706)^T \tag{24}$$

Step 2. Consistency test. Using Equation (8), the maximum eigenvalues are calculated as follows:

$$\lambda_{1\,\text{max}}^- = \frac{1}{5} \sum_{i=1}^n \frac{\left(\Lambda_1^- w_{b_1}^-\right)_i}{w_{b_1 i}^-} = 5.2669, \tag{25}$$

$$\lambda_{1\,\text{max}}^+ = \frac{1}{5} \sum_{i=1}^n \frac{\left(\Lambda_1^+ w_{b_1}^+\right)_i}{w_{b_1 i}^+} = 5.0958 \tag{26}$$

Then, with Equation (9), we obtained

$$CI_1^- = \frac{5.2669 - 5}{5 - 1} = 0.0667, CI_1^+ = \frac{5.0958 - 5}{5 - 1} = 0.0239 \tag{27}$$

Lastly, with Equation (10), we obtained

$$CR_1^- = \frac{CI_1^-}{RI} = \frac{0.0667}{1.12} = 0.0596, CR_1^+ = \frac{CI_1^+}{RI} = \frac{0.0239}{1.12} = 0.0214. \tag{28}$$

Because of $CR_1^- = 0.0596 < 0.1$, $CR_1^+ = 0.0214 < 0.1$, it is considered that matrices $\Lambda_1^-, \Lambda_1^+$ are consistent. Therefore, the derived basic weights from this assessor's comparison judgment matrix are considered to be reliable.

### 4.3.2. Calculating the Weights of Factors $F_1$–$F_5$

According to Equation (12), the upper bound matrix $\Lambda_1^+$ and the lower bound matrix $\Lambda_1^-$, the value $\alpha, \beta$ could be calculated as 0.9958, 0.9979, respectively. So, the final lower/upper bound weights vector from assessor-1 is

$$w_{b_1 L} = \alpha w_{b_1}^- = (0.3033, 0.1160, 0.0706, 0.4004, 0.1056)^T \tag{29}$$

$$w_{b_1 R} = \beta w_{b_1}^+ = (0.4482, 0.1484, 0.0757, 0.2365, 0.0690)^T \tag{30}$$

That is to say, the final interval weights vector from assessor-1 is as Equation (31):

$$w_{b_1} = \left[ w_{b_1 L}, w_{b_1 R} \right] = \quad ([0.3033, 0.4482], [0.1160, 0.1484],$$
$$[0.0706, 0.0757], [0.4004, 0.2365], [0.1056, 0.0690]). \tag{31}$$

Similarly, the values maximum eigenvalues, the consistency ratio (CR), final lower/upper bound weights for interval judgment matrices $\Lambda_i$ ($i = 1, 2, \cdots, 5$) from five assessors could be gained and shown in the following Table 4.

**Table 4.** The weights of factors $F_1$–$F_5$ from five assessors.

| Assessor | $\Lambda$ | $\lambda_{\max}$ | $\alpha/\beta$ | CR | $w_{1L}/w_{1R}$ |
|---|---|---|---|---|---|
| No.1 | $\Lambda_1^-$ | 5.2669 | 0.9779 | 0.0596 | $w_{1L}$=(0.3033, 0.1160, 0.0706, 0.4004, 0.1056) |
|  | $\Lambda_1^+$ | 5.0958 | 0.9958 | 0.0214 | $w_{1R}$= (0.4482, 0.1484, 0.0757, 0.2365, 0.0690) |
| No.2 | $\Lambda_2^-$ | 5.2148 | 0.9896 | 0.0479 | $w_{2L}$=(0.1424, 0.3161, 0.2915, 0.1143, 0.1170) |
|  | $\Lambda_2^+$ | 5.3465 | 0.9813 | 0.0773 | $w_{2R}$= (0.2607, 0.3275, 0.2450, 0.0890, 0.0674) |
| No.3 | $\Lambda_3^-$ | 5.3339 | 0.9733 | 0.0745 | $w_{3L}$= (0.1674, 0.0814, 0.0993, 0.3766, 0.2527) |
|  | $\Lambda_3^+$ | 5.2508 | 0.9775 | 0.0560 | $w_{3R}$= (0.2856, 0.1094, 0.1224, 0.3050, 0.1509) |
| No.4 | $\Lambda_4^-$ | 5.2592 | 0.9804 | 0.0579 | $w_{4L}$= (0.0655, 0.1221, 0.3655, 0.2712, 0.1370) |
|  | $\Lambda_4+$ | 5.4217 | 0.9615 | 0.0941 | $w_{4R}$= (0.1151, 0.1457, 0.3793, 0.2202, 0.1201) |
| No.5 | $\Lambda_5^-$ | 5.3621 | 0.9582 | 0.0808 | $w_{5L}$= (0.2589, 0.2254, 0.0958, 0.1962, 0.1767) |
|  | $\Lambda_5^+$ | 5.4353 | 0.9532 | 0.0972 | $w_{5R}$= (0.3487, 0.2395, 0.0895, 0.1629, 0.1177) |

### 4.4. Assessors' Weights

4.4.1. Calculating the Similarity Coefficient of Assessors' Evaluation

Let $\Lambda_k^- = \begin{pmatrix} a_1^k \\ a_2^k \\ \vdots \\ a_5^k \end{pmatrix}, \Lambda_k^+ = \begin{pmatrix} b_1^k \\ b_2^k \\ \vdots \\ b_5^k \end{pmatrix}$ ($k = 1, 2, \cdots, 5$), according to the proposed method in Section 3.2 and the matrices $\Lambda_1^- \sim \Lambda_5^-$, $\Lambda_1^+ \sim \Lambda_5^+$, and applying Equation (13), the lower similarity matrix and upper similarity matrix are computed, respectively, as follows:

$$P^- = \begin{pmatrix} 1.0000 & 0.4034 & 0.8128 & 0.4633 & 0.7597 \\ 0.4034 & 1.0000 & 0.4399 & 0.7378 & 0.6330 \\ 0.8128 & 0.4399 & 1.0000 & 0.5922 & 0.7727 \\ 0.4633 & 0.7378 & 0.5922 & 1.0000 & 0.4580 \\ 0.7597 & 0.6330 & 0.7727 & 0.4580 & 1.0000 \end{pmatrix}, \tag{32}$$

$$P^+ = \begin{pmatrix} 1.0000 & 0.6265 & 0.8543 & 0.4604 & 0.8527 \\ 0.6265 & 1.0000 & 0.5121 & 0.7013 & 0.7172 \\ 0.8543 & 0.5121 & 1.0000 & 0.6045 & 0.8270 \\ 0.4604 & 0.7013 & 0.6045 & 1.0000 & 0.4921 \\ 0.8527 & 0.7172 & 0.8270 & 0.4921 & 1.0000 \end{pmatrix} \tag{33}$$

By applying Equations (14) and (15), the lower/upper similarity coefficient matrices $s^-/s^+$ could be obtained, respectively, as

$$s^- = (0.2008, 0.1823, 0.2155, 0.1854, 0.2160), \tag{34}$$
$$s^+ = (0.2101, 0.1923, 0.2104, 0.1699, 0.2173) \tag{35}$$

4.4.2. Calculating the Degree of Difference of Assessors' Evaluation and the Weight of Evaluation Assessors

By applying the Equations (16)–(18), the lower/upper degree of difference of five assessors are

$$D^- = (0.2212, 0.2069, 0.1811, 0.2409, 0.1499),$$
$$D^+ = (0.2015, 0.2198, 0.1705, 0.2637, 0.1444) \tag{36}$$

Based on Equation (19) and the data from Equations (34)–(36), the lower/upper weights of the five assessors will, respectively, be

$$\omega_a^- = (0.2040, 0.1886, 0.1845, 0.1835, 0.2394),$$
$$\omega_a^+ = (0.2089, 0.1868, 0.2173, 0.1557, 0.2314) \tag{37}$$

*4.5. Final Weights for Factors $F_1$–$F_5$*

According to the operation rules of interval numbers and Equation (20), the factors' final lower/upper weights could be aggregated as shown in Table 5.

**Table 5.** The final weights for factors $F_1$–$F_5$.

| Factor | Lower Weight | Upper Weight |
|--------|--------------|--------------|
| $F_1$ | 0.1936 | 0.3030 |
| $F_2$ | 0.1747 | 0.1940 |
| $F_3$ | 0.1777 | 0.1679 |
| $F_4$ | 0.2695 | 0.2043 |
| $F_5$ | 0.1577 | 0.1057 |

After normalizing the weights of factors $F_1$–$F_5$, the normalized final weights for factors $F_1$–$F_5$ obtained and listed in Table 6.

**Table 6.** The normalized final weights for factors $F_1$–$F_5$.

| Factor | Lower Weight | Upper Weight |
|--------|--------------|--------------|
| $F_1$ | 0.1990 | 0.3108 |
| $F_2$ | 0.1795 | 0.199 |
| $F_3$ | 0.1826 | 0.1722 |
| $F_4$ | 0.2769 | 0.2095 |
| $F_5$ | 0.162 | 0.1085 |

By repeating the steps in Sections 4.2–4.5, items $I_1 \sim I_{25}$'s weight could be calculated too. It should be noted that the assessors' weights for all items $I_1 \sim I_{25}$ are regarded as the same values, and the interval judgment matrices of each five items from five assessors are listed in Appendix A. Then, the weights of all factors and items in RTOP are obtained as shown in Table 7.

**Table 7.** The weights of all factors and items in RTOP.

| Factor | Lower Weight $(w_{F_i}^-)$ | Upper Weight $(w_{F_i}^+)$ | Items | Lower Weight $(w_{I_i}^-)$ | Upper Weight $(w_{I_i}^+)$ |
|---|---|---|---|---|---|
| $F_1$ | 0.1990 | 0.3108 | $I_1$ | 0.1243 | 0.1717 |
| | | | $I_2$ | 0.2183 | 0.2497 |
| | | | $I_3$ | 0.2604 | 0.2516 |
| | | | $I_4$ | 0.2487 | 0.2128 |
| | | | $I_5$ | 0.1482 | 0.1142 |
| $F_2$ | 0.1795 | 0.1990 | $I_6$ | 0.1282 | 0.1914 |
| | | | $I_7$ | 0.1439 | 0.1768 |
| | | | $I_8$ | 0.3149 | 0.3012 |
| | | | $I_9$ | 0.2101 | 0.1684 |
| | | | $I_{10}$ | 0.2029 | 0.1623 |
| $F_3$ | 0.1826 | 0.1722 | $I_{11}$ | 0.0878 | 0.1460 |
| | | | $I_{12}$ | 0.2063 | 0.2454 |
| | | | $I_{13}$ | 0.2499 | 0.2462 |
| | | | $I_{14}$ | 0.3182 | 0.2549 |
| | | | $I_{15}$ | 0.1378 | 0.1075 |
| $F_4$ | 0.2769 | 0.2095 | $I_{16}$ | 0.2008 | 0.2646 |
| | | | $I_{17}$ | 0.1628 | 0.1938 |
| | | | $I_{18}$ | 0.1574 | 0.1475 |
| | | | $I_{19}$ | 0.2054 | 0.1758 |
| | | | $I_{20}$ | 0.2737 | 0.2183 |
| $F_5$ | 0.1620 | 0.1085 | $I_{21}$ | 0.2002 | 0.2694 |
| | | | $I_{22}$ | 0.2300 | 0.2444 |
| | | | $I_{23}$ | 0.1802 | 0.1816 |
| | | | $I_{24}$ | 0.2209 | 0.1892 |
| | | | $I_{25}$ | 0.1687 | 0.1155 |

## 5. Comprehensive Evaluation with Interval Numbers

### 5.1. Evaluation Standard and Data Collection

As previously discussed, when assessors are invited to evaluate a course under evaluation using RTOP as the instrument, there will be difficulties in giving crisp values to express their judgments or opinions. On the contrary, interval numbers are found to be more suitable and reasonable for assessors to conduct their evaluation. Meanwhile, items' fuzziness and assessors' subjective uncertainties were considered by combining qualitative and quantitative methods when evaluating classroom teaching. That is, assessors not only consider the number of occurrences of the item, but also combine their experience to make semantic judgments. Five assessors are needed to evaluate all items during the given mathematics course observation with uncertain interval numbers. Each item is evaluated on a five-level scale ranged from 0 to 1 (from "never occurred" to "very descriptive" scale [46]). It should also be pointed out here that this behavior does not mean that full marks should be given if it occurs four times. In this case, the total evaluation scores will belong in 0 to 1.

With this evaluation scale, the evaluation values of five assessors were collected and listed in Table 8.

**Table 8.** Evaluation values from five assessors.

| Factors | Items | Evaluation Value ($x_{ij}, i = 1, 2, \cdots, 25, j = 1, 2, \cdots, 5$) | | | | |
| | | Assessor-1 | Assessor-2 | Assessor-3 | Assessor-4 | Assessor-5 |
|---|---|---|---|---|---|---|
| $F_1$ | $I_1$ | [0.6,0.7] | [0.65,0.75] | [0.6,0.8] | [0.7,0.8] | [0.8,0.9] |
| | $I_2$ | [0.7,0.8] | [0.8,0.85] | [0.75,0.85] | [0.7,0.8] | [0.85,0.9] |
| | $I_3$ | [0.65,0.75] | [0.7,0.8] | [0.55,0.6] | [0.5,0.6] | [0.6,0.8] |
| | $I_4$ | [0.65,0.75] | [0.7,0.8] | [0.55,0.6] | [0.5,0.6] | [0.6,0.8] |
| | $I_5$ | [0.4,0.5] | [0.5,0.55] | [0.5,0.6] | [0.6,0.7] | [0.6,0.65] |
| $F_2$ | $I_6$ | [0.8,0.9] | [0.75,0.8] | [0.8,0.85] | [0.75,0.85] | [0.8,0.9] |
| | $I_7$ | [0.6,0.8] | [0.7,0.8] | [0.8,0.85] | [0.8,0.9] | [0.75,0.8] |
| | $I_8$ | [0.8,0.9] | [0.8,0.85] | [0.8,0.9] | [0.9,0.9] | [0.75,0.85] |
| | $I_9$ | [0.5,0.6] | [0.75,0.8] | [0.6,0.7] | [0.65,0.7] | [0.785,0.85] |
| | $I_{10}$ | [0.8.0.9] | [0.8,0.85] | [0.8,0.9] | [0.75,0.8] | [0.75,0.85] |
| $F_3$ | $I_{11}$ | [0.6,0.6] | [0.55,0.7] | [0.6,0.65] | [0.5,0.6] | [0.7,0.75] |
| | $I_{12}$ | [0.65,0.7] | [0.65,0.75] | [0.7,0.75] | [0.8,0.9] | [0.75,0.8] |
| | $I_{13}$ | [0.75,0.8] | [0.7,0.75] | [0.7,0.8] | [0.6,0.8] | [0.75,0.85] |
| | $I_{14}$ | [0.5,0.6] | [0.4,0.5] | [0.6,0.65] | [0.5,0.55] | [0.6,0.7] |
| | $I_{15}$ | [0.7,0.8] | [0.6,0.65] | [0.6,0.7] | [0.7,0.8] | [0.75,0.85] |
| $F_4$ | $I_{16}$ | [0.75,0.9] | [0.7,0.8] | [0.7,0.8] | [0.6,0.8] | [0.7,0.75] |
| | $I_{17}$ | [0.75,0.8] | [0.7,0.8] | [0.75,0.85] | [0.65,0.8] | [0.8,0.85] |
| | $I_{18}$ | [0.5,0.6] | [0.55,0.6] | [0.45,0.5] | [0.6,0.7] | [0.6,0.7] |
| | $I_{19}$ | [0.35,0.4] | [0.5,0.6] | [0.55,0.6] | [0.5,0.6] | [0.5,0.6] |
| | $I_{20}$ | [0.75,0.8] | [0.7,0.8] | [0.65,0.7] | [0.6,0.7] | [0.75,0.8] |
| $F_5$ | $I_{21}$ | [0.8,0.9] | [0.75,0.8] | [0.75,0.85] | [0.8,0.85] | [0.75,0.85] |
| | $I_{22}$ | [0.6,0.7] | [0.7,0.75] | [0.6,0.65] | [0.7,0.8] | [0.7,0.8] |
| | $I_{23}$ | [0.8,0.9] | [0.85,0.9] | [0.75,0.8] | [0.8,0.9] | [0.7,0.8] |
| | $I_{24}$ | [0.6,0.8] | [0.6,0.7] | [0.65,0.7] | [0.75,0.8] | [0.7,0.8] |
| | $I_{25}$ | [0.6,0.7] | [0.5,0.6] | [0.45,0.5] | [0.5,0.6] | [0.6,0.7] |

*5.2. Comprehensive Evaluation*

Step 1. Calculating the average of five assessors' evaluation values using Equation (38).

$$x_i = \left[x_i^-, x_i^+\right] = \sum_{j=1}^{5} x_{ij} \bigg/ 5 \quad (i = 1, 2, \cdots, 25) \tag{38}$$

Step 2. Calculating $F_1$'s comprehensive values, by aggregating the items' evaluation values with its corresponding weights (the same way to obtain $F_2 \sim F_5$'s comprehensive values) as given in Equation (39).

$$y_1 = \left[y_1^-, y_1^+\right] = \left[\sum_{i=1}^{5} w_{I_i}^- x_i^-, \ \sum_{i=1}^{5} w_{I_i}^+ x_i^+, \ \right] (i = 1, 2, \cdots, 5) \tag{39}$$

Step 3. Determining the final comprehensive evaluation score using Equation (40).

$$y = \left[\sum_{i=1}^{5} w_{F_i}^- y_i^-, \ \sum_{i=1}^{5} w_{F_i}^+ y_i^+, \ \right] (i = 1, 2, \cdots, 5) \tag{40}$$

Finally, the comprehensive evaluation results are listed in Table 9.

**Table 9.** Comprehensive evaluation results.

| Factor | Weight ($w_{F_i}$) | Average Evaluation Value ($y_i$) | Aggregated Score ($S_{F_i} = w_{F_i} y_i$) | Total Aggregated Score ($y$) |
|---|---|---|---|---|
| $F_1$ | [0.1990,0.3108] | [0.6135,0.721] | [0.122,0.2241] | |
| $F_2$ | [0.1795,0.1990] | [0.7549,0.8372] | [0.1355,0.1666] | |
| $F_3$ | [0.1826,0.1722] | [0.6310,0.7193] | [0.1152,0.1239] | [0.6565,0.7510] |
| $F_4$ | [0.2769,0.2095] | [0.6257,0.7238] | [0.1733,0.1517] | |
| $F_5$ | [0.1620,0.1085] | [0.6817,0.7816] | [0.1104,0.0847] | |

Therefore, according to the evaluation level in Table 10, the quality of this course is "good".

**Table 10.** Description of the evaluation scale.

| Interval Scale | Evaluation Level | Description |
|---|---|---|
| (0,0.2] | Very poor | The behavior never occurred, the performance is very poor |
| (0.2,0.4] | Poor | The behavior occurred at least once, the performance is poor to describe the lesson |
| (0.4,0.6] | Medium | The behavior occurred more than once, the performance very loosely describes the lesson |
| (0.6,0.8] | Good | The behavior occurred more than two times, the performance fairly descriptive of the lesson |
| (0.8,1] | Very good | The performance extremely descriptive of the lesson |

## 6. Results and Analysis

### 6.1. Results and Analysis of Interval Weights

6.1.1. Ranking for Interval Weights

It is necessary to rank the index's relative weight in an evaluation, which is useful for teachers' teaching preparation, and to improve their teaching quality with a focused goal. However, for any two interval weights $v_k = \left[v_k^-, v_k^+\right]$, $v_l = \left[v_l^-, v_l^+\right]$, there is greater difficulty and complexity in comparison and ranking for interval numbers than crisp numbers. To achieve this goal, several methods including the midpoints of interval numbers [59,60] and possibility-degree [52,61,62] have been proposed in the literature. This study adopts a simple yet effective possibility-degree method in [52] to rank the interval weights.

For any two interval weights, $v_k = \left[v_k^-, v_k^+\right]$, $v_l = \left[v_l^-, v_l^+\right]$, $v_k$ and $v_l$ are put on x-axis and y-axis, respectively. Based on four peaks $\left(v_k^-, v_l^-\right)$, $\left(v_k^+, v_l^-\right)$, $\left(v_k^+, v_l^+\right)$, $\left(v_k^-, v_l^+\right)$, then a rectangle could be formed. The straight line $y = x$ separates the rectangle into two sections marked as $\tilde{A}_1$ and $\tilde{A}_2$. In the area of $\tilde{A}_1$, the points satisfying $y > x$, while in $\tilde{A}_2$, $x > y$. Therefore, the possibility-degree [52] are defined as follows.

Let $v_k = \left[v_k^-, v_k^+\right]$, $v_l = \left[v_l^-, v_l^+\right]$ be any two interval weights, $v_k^- \neq v_k^+$ and $v_l^- \neq v_l^+$. Set $\tilde{A} = \left(v_l^+ - v_l^-\right)\left(v_k^+ - v_k^-\right)$, then the possibility-degree of $v_l \geqslant v_k$ is defined by

$$p(v_l \geq v_k) = \frac{\tilde{A}_1}{\tilde{A}} \tag{41}$$

Likewise, the possibility-degree of $v_k \geqslant v_l$ will be

$$p(v_k \geq v_l) = \frac{\tilde{A}_2}{\tilde{A}} \tag{42}$$

Obviously, $0 \leqslant p(v_l \geqslant v_k) \leqslant 1$, $p(v_l \geqslant v_k) + p(v_k \geqslant v_l) = 1$ and $p(v_k \geqslant v_k) = 0.5$.

Applying Equations (41) and (42), the possibility-degree matrix of factor $F_1$-$F_5$'s interval weights in Table 7 can be computed, namely

$$p = \begin{array}{c} w_{F_1} \\ w_{F_2} \\ w_{F_3} \\ w_{F_4} \\ w_{F_5} \end{array} \left( \begin{array}{ccccc} 0.5 & 1 & 1 & 0.6037 & 1 \\ 0 & 0.5 & 0.9763 & 0 & 1 \\ 0 & 0.0237 & 0.5 & 0 & 1 \\ 0.3963 & 1 & 1 & 0.5 & 1 \\ 0 & 0 & 0 & 0 & 0.5 \end{array} \right) \tag{43}$$

Using the row–column elimination method [52,61], the ranking order is derived as

$$w_{F_1} \succ^{0.6037} w_{F_4} \succ^1 w_{F_2} \succ^{0.9763} w_{F_3} \succ^1 w_{F_5}.$$

Similarly, after comparison with items' weights in Table 7, the items with minimum/maximum weights in each factor could be computed as in Table 11.

**Table 11.** Ranking for interval weights of factors and items.

| Factor | Weight | Ranking | Items of Minimum/Maximum Weight |
|--------|--------|---------|--------------------------------|
| $F_1$ | [0.1990,0.3108] | 1 | $I_5 / I_3$ |
| $F_2$ | [0.1795,0.1990] | 3 | $I_6 / I_8$ |
| $F_3$ | [0.1826,0.1722] | 4 | $I_{11} / I_{14}$ |
| $F_4$ | [0.2769,0.2095] | 2 | $I_{18} / I_{20}$ |
| $F_5$ | [0.1620,0.1085] | 5 | $I_{25} / I_{22}$ |

From Table 11, for all factors, "Lesson design and implementation ($F_1$)" gained the relative highest weight in this course, its weight with had the largest changes, from 0.1989 to 0.3108, followed by "Classroom culture: communicative interactions ($F_4$)" and "content: Propositional Knowledge ($F_2$)", while "Classroom culture: student/teacher relationships ($F_5$)" has the least weight in both lower weight and upper weight.

In each corresponding factor, the items with relative minimum/maximum weight are also listed. For example, in $F_1$, the item "adopt student ideas in teaching($I_5$)" takes the lowest weight, while the item "student exploration preceded formal presentation($I_3$)" takes the highest weight. In $F_2$, the item "The lesson involved fundamental concepts of the subject($I_6$)" takes the lowest weight, the item "The teacher had a solid grasp of the subject matter content inherent in the lesson($I_8$)" takes the highest weight, etc.

6.1.2. Analysis for the Ranking Results

To specific reasons for above ranking results of weights, assessors gave their explanation. For the factor ($F_1$), nearly all assessors stated that "teaching design is the key link of teaching, which not only reflects the teachers' serious and earnest attitude, but also reflects the teachers' grasp and control of the whole class". Actually, this view is supported by a great amount of theoretical literature [63]. Additionally, many teachers, including the author herself, also said they have attached great importance to instructional design in practice teaching. As to why assessors thought the factor "Classroom culture: communicative interactions($F_4$)" needed a relative higher weight, about one-half responded that "Good communication and interaction is the signal of students' response or feedback to the teacher's teaching. Otherwise, the classroom will with dull rather than active atmosphere. In this case, the students' learning enthusiasm and initiative cannot be stimulated at all".

When listing the items with relative minimum weight and relative maximum weight in corresponding factors, almost all assessors replied that they accepted these results and gave their view: "Student exploration preceded formal presentation ($I_3$)" is helpful to stimulate the initiative of learning and the acceptance of teaching content. "Having solid grasp of the subject content ($I_8$)" is the threshold to start this lesson. "Students were reflective about their learning ($I_{14}$)" is the first critical step to their own critical thinking. At the same time, they also expressed the reasons for items with the relative lowest weights. It difficulty for teachers to receive the opinion "The focus and direction of the lesson was often determined by students ($I_5$)", because teachers need to complete the teaching plan and teaching content in a set period of time. Otherwise, the teaching plan will be disrupted or delayed. A similar explanation also appeared in "There was a high proportion of student talk and a significant amount of it occurred between and among students ($I_{18}$)". Therefore, the item $I_5$, $I_{18}$ and others were derived with relatively lower weights.

*6.2. Results and Analysis of Aggregated Scores*

6.2.1. Ranking for Aggregated Scores

The possibility-degree method is re-used to grasp and compare all factors' aggregated scores. According to aggregated score in Table 9, and Equations (41)–(42), the possibility-degree matrix of the factors' aggregated scores are calculated as follows.

$$
P = \begin{matrix} s_{F_1} \\ s_{F_2} \\ s_{F_3} \\ s_{F_4} \\ s_{F_5} \end{matrix} \left( \begin{matrix} 0.5 & 0.7155 & 0.998 & 0.6033 & 1 \\ 0.2845 & 0.5 & 1 & 0.1652 & 1 \\ 0.002 & 0 & 0.5 & 0 & 1 \\ 0.3967 & 0.8348 & 1 & 0.5 & 1 \\ 0 & 0 & 0 & 0 & 0.5 \end{matrix} \right) \tag{44}
$$

Then, the aggregated scores of five factors could be ranked as

$$
s_{F_1} \succ^{0.6033} s_{F_4} \succ^{0.8348} s_{F_2} \succ^{1} s_{F_3} \succ^{1} s_{F_5}.
$$

That is, $F_1$ accounts for the highest proportion of the total aggregated score, followed by $F_4$, $F_2$ and $F_3$, while $F_5$ occupies the least proportion of the total aggregated score. All factors' aggregated scores contributed to the total aggregated score $[0.6565, 0.7510]$, so this course's evaluation level is "good".

6.2.2. Analysis of Total Aggregated Scores

Is the evaluation score of the RTOP tool applicable to mathematics courses? The assessors participating in the evaluation have some questions about the results. After all, when converted into a hundred points system, the class of teacher L obtained a total score between 65.65 and 75.10. This score seemed a bit lower than those evaluation scores that did not use RTOP. In fact, with RTOP as a strict classroom teaching observation tool, the total evaluation score for the all items can range between 0 and 100, but most classes' scores are less than 80 [35,50]. Hence, the class of teacher L has gained an ideal evaluation score in this study. This is consistent with his winning the first prize in the school of "First young teachers' lecture competition".

Additionally, to verify the proposed evaluation method in this paper, we used it to aggregate the evaluation data in [50] and obtained an evaluation score of 67.35–68.62. Compared with the the score of 68.38 in [50], the difference is controlled in 1.5%.

**7. Conclusions**

Due to uncertainties and the fuzziness of humans' judgment, it is considered that interval numbers should be more natural, logical and acceptable than crisp numbers to compare the priority of different indicators or conduct evaluations in teaching quality. Therefore, this study chose a widely used instrument, RTOP, as an evaluation tool, and chose interval numbers and the I-AHP method as the main methods to assess the classroom teaching quality for a course on probability theory and mathematical statistics. Since different assessors have different roles, the assessors' objective weights were also incorporated into the factors' weights. The conclusions can be summarized as follows. Firstly, all factors and items' interval weights were obtained with I-AHP, and the evaluation values were collected with interval numbers. Secondly, a given class's teaching quality was evaluated with a comprehensive fuzzy evaluation method. Thirdly, all the factors' interval weights and interval aggregated scores were ranked with a possibility-degree method. From the results of ranking, teachers could easily observe their class's merits and shortcomings, which is useful for them to continue improving in the future. The results demonstrate that the I-AHP method could overcome the typical problem of uncertainties and ambiguities when interpreting results, and yields a wider evaluation score than conventional AHP, which is more reasonable and acceptable in assessment by participants, especially teachers. The instrument RTOP and the method I-AHP were considered helpful for analyzing challenging in teaching. Furthermore, these evaluation results with interval numbers and the

I-AHP method can contribute to more flexible decision-making to continuously improve the teaching quality for mathematics education in higher education.

**Author Contributions:** Conceptualization, Y.Q., S.R.M.H. and J.S.; investigation, Y.Q.; methodology, Y.Q. and S.R.M.H.; writing—original draft, Y.Q.; writing—review and editing, Y.Q., S.R.M.H. and J.S. All authors have read and agreed to the published version of the manuscript.

**Funding:** The research was funded by the General Program of Natural Funding of Sichuan Province (No. 2021JY018), Scientific Research Project of Neijiang Normal University (18TD08, 2022ZD10.), Scientific Research Project of Neijiang City (No. NJFH20-003)

**Institutional Review Board Statement:** Not applicable.

**Informed Consent Statement:** Not applicable.

**Data Availability Statement:** The data presented in this study are available in article.

**Conflicts of Interest:** The authors declare no conflict of interest.

**Appendix A. Interval Reciprocal Judgment Matrices for Items**

Interval reciprocal judgment matrices for $I_1 - I_5$:

$$
\begin{pmatrix}
[1,1] & [0.5,0.75] & [1/3,0.5] & [0.5,1] & [1,1.5] \\
 & [1,1] & [3,3.5] & [2,2.5] & [3,3.5] \\
 & & [1,1] & [1,1] & [3,3.5] \\
 & & & [1,1] & [2,2.5] \\
 & & & & [1,1]
\end{pmatrix},
$$

$$
\begin{pmatrix}
[1,1] & [0.25,1/3] & [0.2,1/3] & [1/3,0.5] & [0.25,0.5] \\
 & [1,1] & [0.25,0.5] & [0.75,1] & [1.5,2] \\
 & & [1,1] & [3,3.5] & [2,2.5] \\
 & & & [1,1] & [2,2.5] \\
 & & & & [1,1]
\end{pmatrix},
$$

$$
\begin{pmatrix}
[1,1] & [1/3,0.5] & [0.2,0.25] & [1/3,0.5] & [0.8,1] \\
 & [1,1] & [0.5,0.8] & [1/3,0.5] & [0.5,1] \\
 & & [1,1] & [2,2.5] & [3,3.5] \\
 & & & [1,1] & [3,4] \\
 & & & & [1,1]
\end{pmatrix},
$$

$$
\begin{pmatrix}
[1,1] & [0.5,0.6] & [0.5,0.8] & [1/3,2/3] & [1,1.5] \\
 & [1,1] & [2,2.5] & [0.5,2/3] & [1,1.2] \\
 & & [1,1] & [0.25,1/3] & [2,2.5] \\
 & & & [1,1] & [1,1.5] \\
 & & & & [1,1]
\end{pmatrix},
$$

$$
\begin{pmatrix}
[1,1] & [3,3.5] & [2,2.5] & [1,2] & [1,1.5] \\
 & [1,1] & [2,2.5] & [2,3] & [1.5,2] \\
 & & [1,1] & [0.5,1] & [0.5,0.8] \\
 & & & [1,1] & [1.5,2] \\
 & & & & [1,1]
\end{pmatrix}.
$$

Interval reciprocal judgment matrices for $I_6 - I_{10}$:

$$
\begin{pmatrix}
[1,1] & [1/3,0.5] & [0.2,1/3] & [1/3,1] & [0.5,1] \\
 & [1,1] & [1/3,0.5] & [0.5,1] & [1,1.5] \\
 & & [1,1] & [0.5,1] & [2,3] \\
 & & & [1,1] & [3,3.5] \\
 & & & & [1,1]
\end{pmatrix},
$$

$$
\begin{pmatrix}
[1,1] & [2,3] & [1/3,0.5] & [1,1.5] & [1/3,0.5] \\
 & [1,1] & [0.25,0.5] & [1/3,0.5] & [0.25,1/3] \\
 & & [1,1] & [3,4] & [2,3] \\
 & & & [1,1] & [1/3,0.5] \\
 & & & & [1,1]
\end{pmatrix},
$$

$$
\begin{pmatrix}
[1,1] & [3,3.5] & [1/3,0.5] & [0.5,1] & [2,3] \\
 & [1,1] & [1/3,0.8] & [1,1.5] & [2,2.5] \\
 & & [1,1] & [2,2.5] & [3,3.5] \\
 & & & [1,1] & [2,3] \\
 & & & & [1,1]
\end{pmatrix},
$$

$$
\begin{pmatrix}
[1,1] & [0.5,0.6] & [1/3,0.8] & [0.25,2/3] & [2,2.5] \\
 & [1,1] & [1,1.5] & [2,2.5] & [3,3.5] \\
 & & [1,1] & [1,1.5] & [2,2.5] \\
 & & & [1,1] & [2,3] \\
 & & & & [1,1]
\end{pmatrix},
$$

$$
\begin{pmatrix}
[1,1] & [3,4] & [0.5,1] & [1.2,2] & [1/3,0.5] \\
 & [1,1] & [1/3,0.5] & [2/3,0.8] & [0.2,1/3] \\
 & & [1,1] & [2,2] & [0.5,1] \\
 & & & [1,1] & [1/3,0.5] \\
 & & & & [1,1]
\end{pmatrix}.
$$

Interval reciprocal judgment matrices for $I_{11} - I_{15}$:

$$
\begin{pmatrix}
[1,1] & [0.5,0.8] & [3,4] & [1/3,0.5] & [1,2] \\
 & [1,1] & [5,7] & [1.5,2] & [4,4.5] \\
 & & [1,1] & [1/7,1/5] & [0.5,1] \\
 & & & [1,1] & [5,7] \\
 & & & & [1,1]
\end{pmatrix},
$$

$$
\begin{pmatrix}
[1,1] & [1/3,0.5] & [0.2,1/3] & [1/3,0.5] & [0.25,1/3] \\
 & [1,1] & [0.25,0.5] & [0.5,1] & [1.5,2] \\
 & & [1,1] & [3,3.5] & [1.5,2.5] \\
 & & & [1,1] & [2,3] \\
 & & & & [1,1]
\end{pmatrix},
$$

$$
\begin{pmatrix}
[1,1] & [1/3,0.5] & [0.2,0.25] & [1/3,0.5] & [0.8,1] \\
 & [1,1] & [0.5,0.8] & [1/3,0.5] & [1,1] \\
 & & [1,1] & [2,2.5] & [3,3.5] \\
 & & & [1,1] & [3,4] \\
 & & & & [1,1]
\end{pmatrix},
$$

$$
\begin{pmatrix}
[1,1] & [0.5,1] & [1/3,0.8] & [0.25,2/3] & [0.5,1] \\
 & [1,1] & [2,2.5] & [0.5,1] & [1,1.2] \\
 & & [1,1] & [0.25,1/3] & [2,2.5] \\
 & & & [1,1] & [1,1.5] \\
 & & & & [1,1]
\end{pmatrix},
$$

$$
\begin{pmatrix}
[1,1] & [0.5,1] & [1/3,0.5] & [1/3,1] & [0.5,1.5] \\
 & [1,1] & [0.5,1.5] & [0.5,1] & [2,2.5] \\
 & & [1,1] & [1/3,1] & [2,3] \\
 & & & [1,1] & [2,2.5] \\
 & & & & [1,1]
\end{pmatrix}.
$$

Interval reciprocal judgment matrices for $I_{16} - I_{20}$:

$$
\begin{pmatrix}
[1,1] & [2,2.5] & [3,3.5] & [4,5] & [1,1.5] \\
 & [1,1] & [1.2,1.5] & [2,2.5] & [0.5,1] \\
 & & [1,1] & [1.2,1.5] & [0.25,0.5] \\
 & & & [1,1] & [0.2,0.25] \\
 & & & & [1,1]
\end{pmatrix},
$$

$$
\begin{pmatrix}
[1,1] & [1.5,2.5] & [2.5,3] & [3,3.5] & [0.2,0.25] \\
 & [1,1] & [1,1.2] & [1.5,2] & [1/6,0.2] \\
 & & [1,1] & [1.5,2] & [0.2,0.25] \\
 & & & [1,1] & [0.2,1/3] \\
 & & & & [1,1]
\end{pmatrix},
$$

$$
\begin{pmatrix}
[1,1] & [0.25,1/3] & [0.2,0.5] & [0.5,1] & [0.8,1] \\
 & [1,1] & [0.5,1] & [1/3,0.5] & [1.5,2] \\
 & & [1,1] & [1/3,0.5] & [1.5,2] \\
 & & & [1,1] & [0.5,1] \\
 & & & & [1,1]
\end{pmatrix},
$$

$$
\begin{pmatrix}
[1,1] & [0.6,1] & [1,1.5] & [0.5,1] & [1.5,2] \\
 & [1,1] & [0.5,1] & [1/3,0.5] & [1.5,2] \\
 & & [1,1] & [1/3,0.5] & [2,2.5] \\
 & & & [1,1] & [4.5,5] \\
 & & & & [1,1]
\end{pmatrix}.
$$

$$
\begin{pmatrix}
[1,1] & [2.5,3.5] & [2,2.5] & [1,2] & [1,1.5] \\
 & [1,1] & [1.5,2.5] & [2,3] & [1.5,2] \\
 & & [1,1] & [0.5,1] & [0.5,0.5] \\
 & & & [1,1] & [1.5,1.5] \\
 & & & & [1,1]
\end{pmatrix}.
$$

Interval reciprocal judgment matrices for: $I_{21} - I_{25}$

$$
\begin{pmatrix}
[1,1] & [1,1.5] & [1.5,2] & [2,2.5] & [1.5,2] \\
 & [1,1] & [1.2,2] & [3,4] & [2,2.5] \\
 & & [1,1] & [2,2.5] & [1.5,2] \\
 & & & [1,1] & [1,1.5] \\
 & & & & [1,1]
\end{pmatrix},
$$

$$
\begin{pmatrix}
[1,1] & [0.5,1] & [1,1.5] & [1,1.5] & [0.5,1] \\
 & [1,1] & [1,1.2] & [2,3] & [1,1.5] \\
 & & [1,1] & [1.5,2] & [1/3,1] \\
 & & & [1,1] & [1.5,2] \\
 & & & & [1,1]
\end{pmatrix},
$$

$$
\begin{pmatrix}
[1,1] & [0.5,1] & [1.5,2] & [2,2.5] & [5,6] \\
 & [1,1] & [1.2,1.5] & [3,3.5] & [1.5,2] \\
 & & [1,1] & [2,3] & [1,2] \\
 & & & [1,1] & [2,2.5] \\
 & & & & [1,1]
\end{pmatrix},
$$

$$
\begin{pmatrix}
[1,1] & [0.5,1] & [1/3,0.5] & [0.25,0.5] & [1,1.5] \\
 & [1,1] & [0.5,1] & [0.5,2/3] & [1,2] \\
 & & [1,1] & [0.25,0.25] & [2,3] \\
 & & & [1,1] & [1,2] \\
 & & & & [1,1]
\end{pmatrix},
$$

$$
\begin{pmatrix}
[1,1] & [2.5,3.5] & [2,2.5] & [0.5,1] & [1,1.5] \\
 & [1,1] & [2,2.5] & [1/3,0.5] & [1.5,2] \\
 & & [1,1] & [0.5,1] & [0.5,0.8] \\
 & & & [1,1] & [1.5,2] \\
 & & & & [1,1]
\end{pmatrix}.
$$

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
