# Peer review of "An Interval AHP Technique for Classroom Teaching Quality Evaluation"

_education, doi:10.3390/educsci12110736_

Round 1
Reviewer 1 Report
The reviewed study is a well-thought-out and complete whole concerning the instrument RTOP and the method I-AHP, which were considered helpful for analyzing challenging in teaching. Further, these evaluation results with interval numbers and I-AHP method can contribute more flexible decision-making to continuously improve teaching quality for higher education.
Author Response
Dear Reviewer,thank you for your valuable comments.
Reviewer 2 Report
Hierarchical evaluation structure is applying for the classroom teaching quality evaluation through the analytic hierarchy process measuring the uncertain factors associated to the values of the scale of Importance constructing comparative judgement matrices producing consistent results. Positive, real intervals supported the weighted matrices determining assessors and criteria for the comprehensive evaluation.
The explanation is theoretical with some hint solving numerical examples for the applied probabilistic and statistic topic. Reformed Teaching Observation Protocol is the strength of this method because it enhances the reliability and validity of modern tests on series of data.
It is recommended to add a Section for one completed and computed example applying the I-AHP at practises using one expressed software and to comment more expertise and performance tools.
The employed methods must project to an example or more verifying the validity of the applied contexts.
Reviewer 3 Report
In abstract, there is a repeated sentence in lines 24 and 025
Author Response
Thank you for suggestions. We revised the mistake and deleted the repeated sentence in Paragraph 1, Section I.